# Mangcuo Lake in Hengduan Mountains: An Important Alpine Breeding and Stopover Site along Central Asian Flyway

**DOI:** 10.3390/ani13071139

**Published:** 2023-03-23

**Authors:** Fang Wang, Yongbing Yang, Gang Song, Xiaojuan Shi, Bu Pu, Le Yang

**Affiliations:** 1School of Science, Tibet University, Lhasa 850001, China; 2Tibet Plateau Institute of Biology, Lhasa 850008, China; 3Institute of Zoology, Chinese Academy of Sciences, Beijing 100101, China; 4School of Economics and Management, Tibet University, Lhasa 850001, China

**Keywords:** Hengduan Mountains, Mangcuo Lake, migration stopover site, Central Asian flyway, waterbirds

## Abstract

**Simple Summary:**

The Hengduan Mountain area in southwest China is an important migratory channel for migratory birds in the Central Asian flyway. The lakes along the way have different ecological functions for the life histories of waterbirds. In order to determine the ecological functions of high-altitude lakes, we selected Mangcuo Lake in Markam County to investigate the waterbird community in four seasons. The results show that Mangcuo Lake provides a breeding habitat for nine species of birds and also supports excellent numbers of birds migrating in the spring and autumn. In view of the role of Mangcuo Lake in the migration route, we propose upgrading the protection level of the Mangcuo Lake.

**Abstract:**

The stopovers provide food and habitat for migratory birds and therefore play an important role in facilitating the birds’ completion of their migration. The Hengduan Mountains, which have a complex topography, are located in a large corridor of the Central Asian migratory flyway, so the lakes along the Hengduan Mountains are important for waterbird migration. The existing research on lakes in the Hengduan Mountain area is mostly concentrated in the central and southern parts of the mountains, which proves that many lakes are wintering grounds for migrating birds. We wonder whether the ecological functions of lakes will change more with further elevation. With this question, we conducted four surveys for the seasonal bird survey in Mangcuo Lake, which is located in the northwest of the Hengduan Mountains, in Markam County of Qamdo City, between October 2019 and July 2020. We recorded a total of 6109 birds from 20 species of waterbirds, including 20 species of migratory waterbirds, accounting for 100% of all bird species. The diversity and richness of waterbirds in Mangtso Lake is shown as spring > autumn ≥, summer > winter, with no waterbirds in winter. The black-necked grebe (*Podiceps nigricollis*), great crested grebe (*Podiceps cristatus*), bar-headed goose (*Anser indicus*), and ruddy shelduck (*Tadorna ferruginea*) were the dominant species in the waterbird community. The highest number of waterbird species and total individuals were found in the transition zone between the marsh wetlands and lakes, and the number of waterbird species differed significantly among habitats (X^2^ = 14.405, *p* = 0.000), with habitat complexity being an important factor influencing waterfowl abundance and distribution. The IUCN Red-listed species recorded include the black-necked crane (*Grus nigricollis*), painted stork (*Mycteria leucocephala*), and common pochard (*Aythya ferina*). By comparing with other lakes in Hengduan Mountain, we found that the ecological functions of the plateau lakes in the Hengduan Mountains, to support the life histories of migrating waterbirds, are gradually transforming as the altitude rises, and can be divided into approximately three levels, with the first level of Qionghai, Chenghai, Erhai, and Jianhu at altitudes of 1500–2200 m being the most important ecological function in terms of providing wintering grounds for migrating birds. The second layer, at an altitude of 2400–3300 m, includes Lashihai, Lugu Lake, and Napahai, which are not only wintering wetlands for migratory birds but also important stopover sites. The third layer of Mangcuo Lake, which is above 4000 m above sea level, provides a breeding ground for some migratory waterbirds in summer and a migratory resting place for migratory waterbirds in spring and autumn. We advocate for the importance of Mangcuo Lake in the alpine region along the central Asian flyway, as well as emerging nature conservation action that was previously neglected.

## 1. Introduction

Wetlands are among the most productive ecosystems on Earth, and waterbirds are sensitive to the changes in wetland environments because they depend on a number of wetlands along their migration routes [1,2,3]; environmental changes can affect waterbird populations directly or indirectly, and wetland suitability is a major determinant of the waterbird population’s dispersal and distribution [4,5,6]. The species diversity and population abundance of birds have long been considered as bio-indicators of environmental change in wetlands [7]. Therefore, understanding the characteristics of the avian community structure is essential to grasping the biodiversity of wetlands and their ecological properties.

Migration is an important behavioral strategy in birds to avoid adverse conditions, either in the breeding areas or wintering areas [8]. The migration routes are usually composed of fixed wintering grounds, stopovers, and breeding grounds [9]. Wetland habitats are ecologically important for waterbird migration, especially the series of available stopover wetlands distributed along the migration route, which are the basis for successful waterbird migration that provide food and habitat for different populations [10,11]. According to Ma et al. [2], stopover places are vital for migratory waterbirds to complete their life cycle. They are the key links that connect breeding and wintering habitats, as well as areas where waterbirds refuel with food and gain energy during long-distance migration [12,13]. Studies have shown that the loss of stopovers is one of the major reasons for migratory birds’ population decline globally [14,15].

Southwest China is located in a large corridor of the Central Asian migratory flyway, where the Hengduan Mountains have a complex topography and are among the 25 global biodiversity hotspots [16,17]. The Hengduan Mountains are distributed with lakes at different altitudes, and the lakes are above 1500 m altitude. Most of the lakes are deep-water lakes, characterized by low temperature, high ultraviolet radiation, and neutrophilic or depleted water quality [18,19]. A review of the literature shows that the ecological function of the highland lakes in the Hengduan Mountains in supporting the life histories of migrating waterbirds is gradually transforming as the altitude rises. Qionghai, Chenghai, Erhai, and Jianhu are distributed at an altitude of 1500–2200 m, and the most important ecological function of these lakes is to provide wintering grounds for migratory birds from Central Asia–India [20,21,22,23]. Lashihai, Lugu Lake, and Napa Sea, which are all 2400–3300 m above sea level, are not only wintering wetlands for migratory birds but also important stopover sites [24,25,26]. The Mangcuo Lake is located in the northwest part of the Hengduan Mountains at an altitude of over 4000 m. During the first comprehensive scientific expedition to the Tibetan Plateau in the 1960s and 1980s, the location of Mangcuo Lake was remote and precarious, the roads were not cleared, and no bird data were available for the area [27]. In the decades since, there have been no reports on the status of bird resources in the area. Our purpose was to investigate the function of Mangcuo Lake’s role in the Central Asia–India migration corridor, and we therefore, carried out a four-season bird survey study of the region.

## 2. Study Area and Methods

### 2.1. Study Area

Mangcuo Lake is located in Markam County, Qamdo City, Tibet Autonomous Region (TAR), China, which is located in the southeast of Markam County and belongs to the Jinsha River Basin (98°47′10″~98°53′41″ E, 29°27′2″~29°38′26″ N, 4313 m.a.s.l). Mangcuo Lake is a high mountain tectonic fault trap lake (Figure 1). The water body of the lake spans 18 km^2^ in total, with a maximum depth of 21 m. It is the largest alpine freshwater lake in the Hengduan Mountains, measuring above 4000 m [19]. The climate in this area is dominated by semi-humid monsoons, with rainy and humid summers and dry and cold winters. The average annual temperature is 3.6℃, and the average annual precipitation is 541 mm [28].

Mangcuo Lake is a typical highland wetland ecosystem; the vegetation is dominated by an alpine subarctic shrub meadow. The meadow vegetation includes Kobresia pygmaea, Polygonum macrophyllum, and Blysmus sinocompressus; the aquatic vegetation is Potamogeton pectinatus of the family Potamogetonaceae; and the marsh vegetation is Triglochin maritimum and Hippuris vulgaris. It provides food and habitats for birds and other highland wildlife. The surveys were conducted in two representative habitats in the lake area: a relatively complex marsh wetland consisting of a transition zone of herbaceous marshes and large lakes, and a relatively homogeneous lake habitat dominated by highland freshwater lakes.

### 2.2. Method

#### 2.2.1. Bird Survey

From November 2019 to July 2020, 4 field surveys on the waterbird community of Mangcuo Lake were conducted using the zonal direct count method. Surveys were conducted in November (autumn), December (winter), April (spring), and July (summer), once in each season. The surveys were carried out by two teams of 2–3 persons, each in clear weather conditions without fog or strong wind. Two survey areas were set up according to the topography, openness, and traffic conditions of the lake area. Area 1 was located in the transition area between mudflat and shoals in the northeast of the lake, and area 2 was located in a single pond in the southeast (Figure 2). We used a Leica 82 mm telescope to observe the birds, positioned with Garmin GPS 62sc. We identified and recorded the species, numbers, and habitat types of the waterbirds. Smaller groups of birds were counted directly, and larger groups were estimated based on the density of birds within different visual field patches and the number of patches.

Classification, scientific, and Chinese names of bird species were based on the book ‘List of the Classification and Distribution of Birds in China’ (3rd ed.), the Zoogeography of China and Birds of the Hengduan Mountain Region [27,29,30]. The conservation status of each species was based on the International Union for Conservation of Nature’s (IUCN) Red List (https://www.iucnredlist.org/ accessed on 1 December 2022) and China’s 2021 State Council Wildlife List, which are used to classify species as endangered or protected [31].

#### 2.2.2. Data Analysis

We applied the Shannon–Wiener (H) diversity index (H)
$$H=-\sum \left(Pi\right)\ln\left(Pi\right)$$
to determine the alpha diversity of birds. The Pi is the proportion of species i to the total species number. We used the Pielou evenness index (J)
$$J=H^{\prime }/H_{max}$$
to evaluate the evenness of the bird community. The S in the formula is the number of bird species in the survey area. The Sorenson similarity index (C)
$$C=2c∕\left(a+b\right)$$
is used to analyze the similarity of bird communities among different seasons for beta diversity, where a is the number of species found in season a, b is the number of species found in season b, and c is the number of common species found in seasons a and b. We define the C value 0–0.25 as very dissimilar, moderately dissimilar for 0.25–0.50, moderately similar for 0.50–0.75, and very similar for 0.75–1.00. Greater values of beta diversity show greater similarity, while smaller values indicate less similarity. The Berger–Parker dominance index (I)
I=Ni∕N
is used for species dominance in different seasons, where Ni is the number of individuals of species I and N is the total number of individuals of all species. The number classes are divided into three classes: dominant species, common species, and rare or occasional species, and defined ≥ 0.05 as dominant species, 0.005 ≤ I < 0.05 as common species, and I < 0.005 as rare or occasional species.

The Shapiro test was used to test the normality of the data. If they conformed to the normal distribution, one-way ANOVA was used for analysis; if they did not conform to the normal distribution, Kruskal–Wallis H was used for analysis, or the data were transformed into species presence/absence data, i.e., species detected at the sample sites were recorded as 1 and those not detected were recorded as 0, and binomial analysis was performed. These methods were used to test for seasonal and annual variability among waterbirds in different habitats. The data were analyzed using R 4.2.1 software with a significance criterion of *p* < 0.05.

## 3. Results

### 3.1. Waterbird Species Composition

The four-season survey recorded 6109 individuals in total, belonging to 20 species in 6 orders and 8 families. The black-necked grebe (I = 63%), the great crested grebe (I = 10%), the bar-headed goose (I = 11%), and the ruddy shelduck (I = 9%) were the dominant species in the waterbird community. The gadwall (*Mareca strepera*), tufted duck (*Aythya fuligula*), and Eurasian coot (*Fulica atra*) are common species in the waterbird community of the region. The number of rare or occasional species was relatively high, with 13 species accounting for 65% of the total number of waterbird species, mainly ducks, storks, and herons. Among the 20 species of waterbirds recorded, 11 species were migrants (55%), and nine species were summer visitors (45%); no waterbird was recorded in winter, indicating that migrants are the majority of waterbirds in Mangcuo Lake (Appendix A).

The Mangcuo Lake is relatively rich in protected birds. There were several species on the IUCN Red List, such as the common pochard (*Aythya ferina*) (VU), black-necked crane, and painted stork (NT). In addition to these, there are also Chinese national protected birds: black stork (*Ciconia nigra*) and black-necked grebe.

### 3.2. Waterbird Diversity and Seasonal Dynamics

A total of 15 species of waterbirds were recorded in Mangcuo Lake in the spring, accounting for 75% of the total number of waterbird species with 4210 individuals. The largest number of species was geese and ducks, with seven species accounting for 47% of the total number of waterbird species in spring; the largest number of species was grebes, with 3668 individuals, accounting for 87% of the total number of waterbirds in spring; and the dominant species in spring were the ruddy shelduck and black-necked grebe. A total of nine species of waterbirds were recorded in the summer, accounting for 45% of the total number of waterbird species and 801 individuals. The species with the largest number of individuals were geese and ducks, with 566 individuals, accounting for 71% of the total number of waterbirds in summer. The dominant species in summer were bar-headed goose, ruddy shelduck, and black-necked grebe. In autumn, nine species of waterbirds were recorded, accounting for 45% of the total number of waterbird species, with 1098 individuals. The largest number of species was geese and ducks, with four species accounting for 44% of the total number of waterbird species in autumn; the largest number of species was grebes, with 625 individuals, accounting for 57% of the total number of waterbirds in the autumn; and the dominant species in the autumn were the ruddy shelduck, tufted duck, great crested grebe, and Eurasian coot. In winter, the lake was frozen, and no waterbirds were recorded.

Among the four seasons, the highest number of waterbird species and individuals was recorded in the spring. The number of waterbird species ranged from high to low in the spring, summer, autumn, and winter, and the number of individuals ranged from high to low in the spring, autumn, summer, and winter. The Shannon–Wiener diversity index and the Pielou index were used to evaluate the alpha diversity of the waterbird community in Mangcuo Lake, and the diversity of waterbirds throughout the year was autumn > summer > spring > winter, with large differences in waterbird diversity between the seasons and no waterbirds in winter (Table 1). In terms of the seasonal dynamics, the species number and abundance were higher in the spring and autumn than in the summer season. The number of species and waterbirds in the migration period accounted for 90% and 87% of the waterbird in reserve, respectively.

The spring and summer seasons had the highest species similarity at 0.583, followed by the spring and autumn seasons at 0.500, while the remaining seasons had a species similarity that was less than 0.500 (Table 2). There are distinct seasonal differences in the composition of bird communities, with few common species and minimal community similarity.

### 3.3. Community Composition of Waterbirds in Different Habitats

In the two survey sample areas, the number of waterbird species recorded in Sample Area 1 was 20, and 8 species of waterbirds were recorded in Sample Area 2 (Table 3). Kruskal-Wallis H analysis showed that the waterbird populations in different habitats did not differ significantly across the seasons, but they differed significantly in terms of the annual numbers (w = 317, *p* = 0.001). A binomial analysis showed that the number of waterbird species differed significantly (X^2^ = 10.025, *p* = 0.002) in both the transition zone of marsh wetlands and lakes and large lake habitats in spring, and not significantly in summer and autumn. The number of species differed significantly between the two habitats on an annual basis (X^2^ = 14.405, *p* = 0.000), with a 100% probability of detecting species in the transition zone of the marsh wetlands sample area and a 40% probability of detecting species in the lake habitat sample area.

## 4. Discussion

### 4.1. Bird Species Composition

Our survey results indicate that the waterbird community of Mangcuo Lake had large differences in the different seasons, with migratory birds accounting for 87% of the total number of birds in spring and autumn, and no waterbirds in winter. Most species of waterbirds in Mangcuo Lake belong to occasional species, and Mangcuo Lake has great potential to discover more migratory occasional species through long-term observations. In terms of numbers, waterbirds were the most abundant in spring, with 4210 birds, while autumn had the highest diversity of waterbirds, with a Shannon-Weiner index of 1.283. It is worth noting that the highest abundance of waterbirds was observed in spring only because of the predominance of the black-necked grebe, resulting in a low diversity calculation for that season.

Waterbirds are extremely sensitive to changes in wetland habitat quality and structure, and their distribution, abundance, and diversity are also closely related to environmental factors such as wetland area, vegetation abundance, and water depth to a certain extent [32]. The survey found that waterbirds in Mangcuo Lake had large differences in species and numbers in the transition zone between the marsh wetland and the lake and in the lake habitat and that habitat complexity was an important factor affecting the number and distribution of birds. The transition zone of marshes and lakes is relatively complex in habitat and rich in aquatic plants, invertebrates, and other resources, making it an important food source for waterbirds, and the lake is shallow and suitable for both wading birds and shallow-water wading birds. The large lakes have a single habitat with few phytoplankton and deeper water than most waterbirds feed at, mainly for deep-water species of wading birds.

The documented migration route of the black-necked grebe is to migrate north from its southern overwintering grounds to breed in Inner Mongolia, northeast China, and northern Xinjiang in April each year and to overwinter in the southeast and south China coasts and southwest China starting in late October [33]. In our study, 3660 black-necked grebes were recorded in Mangcuo Lake in April 2020, while 200 breeding individuals were subsequently observed in July. According to the results of the study, it can be assumed that the black-necked grebe population at Mangcuo Lake is likely to migrate northward from its wintering wetlands in southwestern China in April, and after a stopover at Mangcuo Lake, the majority of individuals continue to go further north, while a small number of pairs breed in Mangcuo Lake. There are no relevant literature reports on the migration studies of black-necked grebes in Tibet, China, and Mangcuo Lake; therefore, it is a good study area.

### 4.2. Mangcuo Lake Is Migratory Stopover Site 

Southwest China’s wetlands, which are located in the Central Asian migration zone of the global migratory corridor and are key wintering grounds and stopovers for many wetland waterbirds, play an important role in the global migration of migratory birds [16]. There are numerous lakes in the Hengduan Mountains (Figure 3), such as the Erhai, Chenghai, Lugu Lake, Qionghai, Napahai, Lashihai, and Jianhu, constituting recognized waterbird distribution habitats [19]. There are some differences between the eco-climate of Mangcuo Lake and the Hengduan Lake Group. Mangcuo Lake is located in the alpine zone, surrounded by mountains on all sides, and is the only alpine freshwater lake with an area of more than 10 km^2^ above 4000 m above sea level in the Hengduan Mountains [19], and it is far away from the lake group, and the distance from the nearest Napa Sea is more than 200 km in a straight line. Mangcuo Lake is a seasonal highland lake with a cold winter climate, with temperatures as low as −10° from December to March when the lake surface freezes. The following April to November, the temperature warms up, and the lake has habitat types such as meadows, shallow water marshes, and open water surfaces. The existing research on lakes in the Hengduan Mountain area is mostly concentrated in the central and southern part of the Mountains, which proves that many lakes are wintering grounds for migrating birds. With the increase in altitude in the north, the ecological functions of some lakes become migratory resting places. The ecological functions of Mangcuo Lake and other lakes in the Hengduan Mountains to support migratory birds are also different through the four-season survey of waterbirds in Mangcao Lake (Table 4). There are no wintering waterbirds in Mangcuo Lake, but there are relatively more species and numbers of migratory waterbirds in spring and autumn, and some geese and grebes breed here in summer.

In the migratory pathway of migratory birds in the Hengduan Mountain system, a series of wetlands are given different ecological functions, and with the elevation, the ecological functions of the lakes in the Hengduan Plateau supporting the life history of migratory waterbirds are gradually transformed about three levels; the first level of Qionghai, Chenghai, Erhai, and Jianhu at an elevation of 1500–2200 m is the most important ecological function of providing important wintering grounds for wintering waterbirds (especially geese and ducks) [20,21,22,23]. The second layer of Lashihai, Lugu Lake, and Napa Sea at 2400–3300 m above sea level are not only wintering wetlands for migratory birds but also important stopover sites [24,25,26]. The third layer, Mangcuo Lake, at an altitude of 4000 m above sea level, provides a breeding ground for some migratory waterbirds in summer and a stopover for migratory waterbirds in spring and autumn. In highland regions, a particular wetland would not be able to supply the dietary demands of all waterbirds due to food resource limitations. Benefiting from the connectivity of migration channels, some birds with high food demands can move around between adjacent wetlands for food [34,35], which may be the reason for the similar ecological functions of adjacent wetlands in the Hengduan Mountains. Migrating in mountainous regions with an altitude of more than 4000 m presents migratory birds with numerous obstacles, including low oxygen levels, low temperatures, and a harsh climate. At this point, Mangcuo Lake, as a place for migratory birds to replenish energy, rest, and adapt to hypoxia, is of great significance to migratory water birds [10].

### 4.3. The Protection Value of Mangcuo Schould Be Improved

Although Mangcuo Lake is an important alpine lake in the northwestern part of the Hengduan Mountains [19], its significant value as a migratory resting place for birds is underestimated. The area around Mangcuo Lake was classified as a county-level nature reserve in 1985 [36]. There are numerous rare and endangered species inhabiting the Mangcuo Lake nature reserve. The black stork recorded in Mangcao Lake in this study is one of the important arguments to confirm the distribution of black storks in Tibet, as there is no documented information on the distribution of black storks in the TAR of China before [37]. The Painted Stork recorded in Mangcuo Lake during the same period is the most recent observation of this species in TAR in the past six decades [38]. In addition to the waterbird survey, we found that the area was also active with Lammergeier (*Gypaetus barbatus*), Himalayan vultures (*Gyps himalayensis*), Tibetan antelope (*Procapra picticaudata*), Tibetan foxes (*Vulpes ferrilata*), wolves (*Canis lupus*), and other rare and endangered wildlife.

Owing to habitat loss and human disturbance, the population of the black-necked grebe has become scarce in China and has been listed as a secondary-priority species for conservation in China in 2021 [31]. The black-necked grebe, which was identified as a new record for Tibetan birds in 2013, recorded 3600 individuals during the migration season in Mangcuo Lake [39]. All of these occurrence records show that Mangcuo Lake is a critical link in the Hengduan Mountains’ migratory path, with significant conservation significance for waterbirds and other rare bird species migrating northward through the alpine area. 

Unfortunately, the water area of Mangcuo Lake has shrunk in recent years [28], owing to warming and drying. Lake vegetation is extremely sensitive to water changes, and the decline of the lake’s water level will cause the distribution area of diverse vegetation communities to retreat, increasing the risk of habitat and food resource loss for the waterbirds. The Mangcuo Lake area is located in an alpine region with a fragile ecological environment, while socioeconomic development and conservation represent the primary conflict. We recommend the formation of a higher-quality nature reserve or a national park in the region to further balance the development and protection, scientific planning, and coordinated development so as to sustain the important ecological role of Mangcuo Lake in supporting the migration of birds.

## 5. Conclusions

(1)The four-season survey recorded 6109 individuals in total, belonging to 20 species in 6 orders and 8 families. The waterbird community of Mangcuo Lake had large differences between different seasons, with migratory birds accounting for 87% of the total number of birds in spring and autumn and no waterbirds in winter. All the waterbirds staying in Mangcuo Lake were migratory birds, and geese, ducks, and grebes were the main migratory waterbirds.(2)We found that the ecological functions of the plateau lakes in the Hengduan Mountains to support the life histories of migrating waterbirds are gradually transforming as the altitude rises and can be divided into approximately three levels, with the first level of Qionghai, Chenghai, Erhai, and Jianhu at altitudes of 1500–2200 m being the most important ecological function of providing wintering grounds for migrating birds. The second layer at an altitude of 2400–3300 m, Lashihai, Lugu Lake, and Napahai, are not only wintering wetlands for migratory birds but also important stopover sites. The third layer of Mangcuo Lake, which is 4000 m above sea level, provides a breeding ground for some migratory waterbirds in summer and a migratory resting place for migratory waterbirds in spring and autumn.(3)The protection value of Mangcuo should be improved. We advocate for the formation of a higher-quality nature reserve or national park in the region to sustain the important ecological of Mangcuo Lake role in supporting the migration of birds.

## Figures and Tables

**Figure 1 animals-13-01139-f001:**
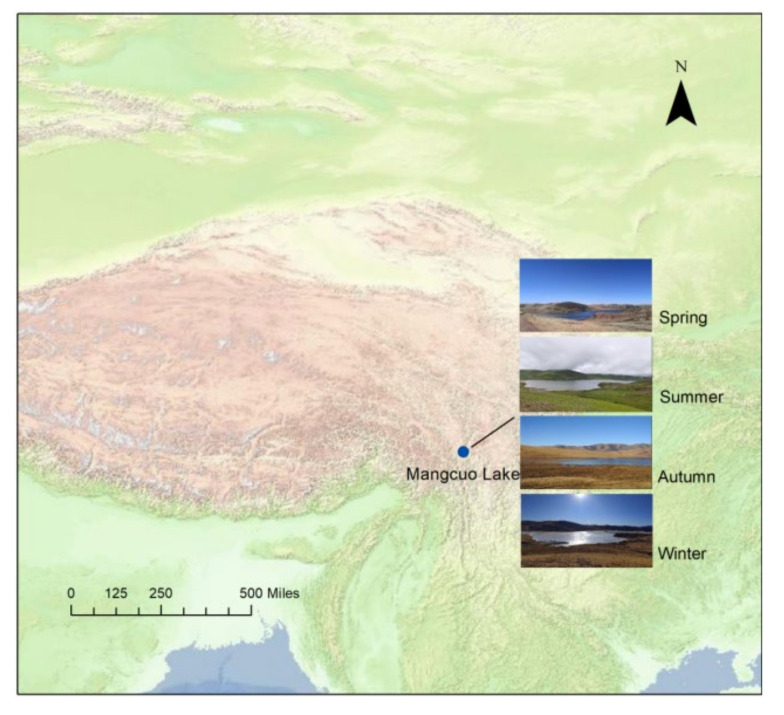
The location of Mangcuo Lake and the real view of the four seasons.

**Figure 2 animals-13-01139-f002:**
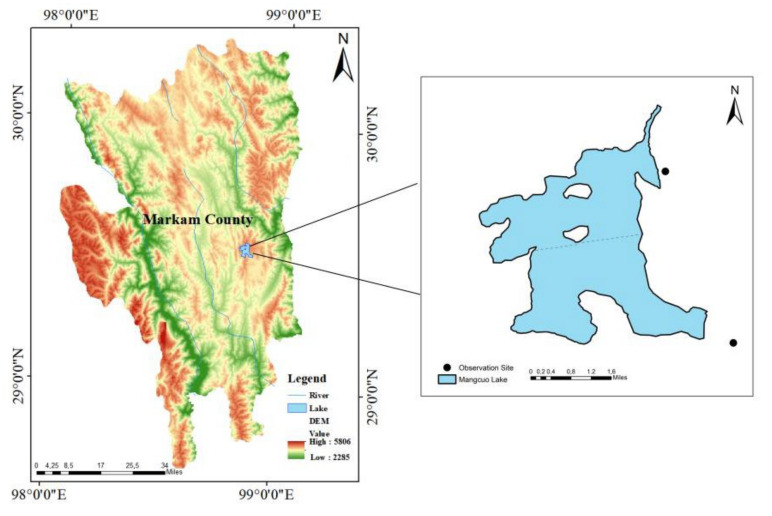
Observation sites and zoning at Mangcuo Lake.

**Figure 3 animals-13-01139-f003:**
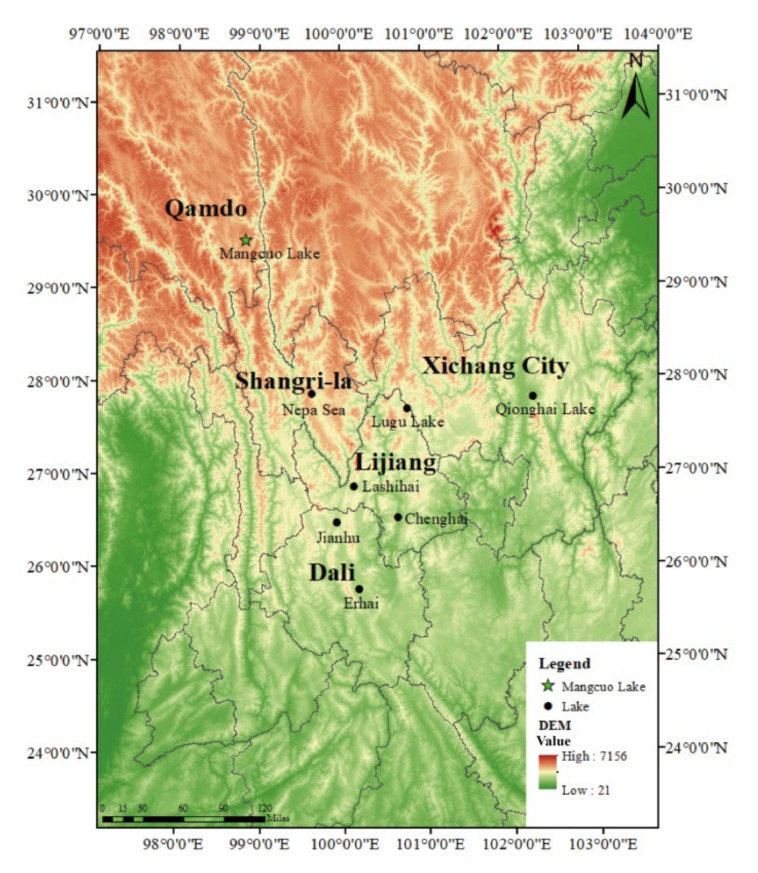
The relative position of Mangcuo Lake and the Hengduan Mountain Lake group.

**Table 1 animals-13-01139-t001:** Seasonal dynamics of bird diversity in Mangcuo Lake.

	Spring	Summer	Autumn	Winter
Species	15	9	9	0
Number of individuals	4210	801	1098	0
Shannon-Wiener index	0.594	1.087	1.283	0
Pielou evenness index	0.219	0.495	0.584	0

**Table 2 animals-13-01139-t002:** Bird community similarity coefficient and number of common species in each season.

Season	Spring	Summer	Autumn
**Spring**		7	6
**Summer**	0.5833		4
**Autumn**	0.5000	0.4444	

Note: The data in the lower left corner are the similarity coefficients, and the data in the upper right corner are the number of common species.

**Table 3 animals-13-01139-t003:** Distribution of birds in different areas.

Sample Area	Spring		Summer		Autumn		Annual	
	Species	Individuals	Species	Individuals	Species	Individuals	Species	Individuals
Sample area 1	15	4039	9	782	9	435	20	5341
Sample area 2	4	171	4	19	5	663	8	853

**Table 4 animals-13-01139-t004:** Relative position of Mangcuo Lake to the lake group.

Lake	Mangcuo Lake	Napa Sea	Lugu Lake	Lashihai	Jianhu	Qionghai	Chenghai	Erhai
Ecological function	Breeding ground and stopovers sites	Wintering wetlands and stopover sites	Wintering wetlands and stopover sites	Wintering wetlands and stopover sites	Wintering wetlands	Wintering wetlands	Wintering wetlands	Wintering wetlands
Azimuth/°	0	South-East66.7°	South-East46.4°	South-East66.1°	South-East72.0°	South-East30.9°	South-East61.6°	South-East72.8°
Distance/km	0	200.78	278.24	321.93	355.00	389.30	377.599	437.29
Area/km^2^	18.00	31.25	51.30	14.43	7.50	31.00	76.9	249.40
altitude/m	4313	3278	2672	2432	2203	1500	1503	1938
Lake contour	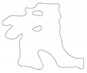	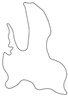	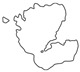	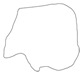	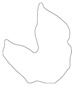	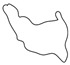	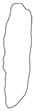	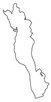

## Data Availability

Not applicable.

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
