# Peer review of "Mangcuo Lake in Hengduan Mountains: An Important Alpine Breeding and Stopover Site along Central Asian Flyway"

_animals, 2023, doi:10.3390/ani13071139_

Round 1

Reviewer 1 Report

The article presents the results of 4 years monitoring of migratory birds in Mangucuo Lake, Hengduan Mountains. The Manguco Lake is an important area on the migration routes of birds in south Asia. 

The article needs many improvements. 

The title is too long

The puropse of the work and the research hypothesis are missing.

The methodology are too general, requires details on number, and timing of counts. Statistical analysis is completely missing. The results lack raw data from individual years and seasons, giving an aggregate number of individuals from 4 years does not provide for analysis and conclusions. 

In the Results section there are elements of methodology and disscusion.  

Discussion needs to be shortened by parts not related to the Results. 

The way the data was collected and compiled does not support the concluding postulate. 

Author Response

Response to Reviewer 1 Comments

Point 1:The title is too long

Response 1: We revised the title into “Mangcuo Lake in Hengduan Mountains is an important alpine breeding and stopover site along the central Asian flyway”.

Point 2:The puropse of the work and the research hypothesis are missing.

Response 2: Thanks for this comment, we modified the introduction section and addressed our aim as follows:

“Our purpose was to investigate the role of Mangcuo Lake in the Central Asia-India migration corridor, and we therefore carried out a four-season bird survey study of the region.”

Point 3:The methodology are too general, requires details on number, and timing of counts. Statistical analysis is completely missing. The results lack raw data from individual years and seasons, giving an aggregate number of individuals from 4 years does not provide for analysis and conclusions.

Response 3: Thanks to your comments, we have reorganized this section with the necessary details supplement.

“From November 2019 to July 2020, four field surveys on the waterbird community of Mangcuo Lake were conducted using the zonal direct count method. Surveys were conducted in November (autumn), December (winter), April (spring), and July (summer), once in each season. The surveys were carried out by two teams of 2-3 persons each in clear weather condition without fog or strong wind. Two survey areas were set up according to the topography, openness, and traffic conditions of the lake area. The area 1 was located in the transition area between mudflat and shoals in the northeast of the lake, and the area 2 was located in a single pond in the southeast. We used a Leica 82 mm telescope to observe the birds, positioned with Garmin GPS 62sc. We identified and recorded the species, numbers and habitat types of the waterbirds. Smaller groups of birds were counted directly, and larger groups were estimated based on the density of birds within different visual field patches and the number of patches.”

Point 4:In the Results section there are elements of methodology and disscusion.  

Response 4: Accepted, we reorganized the results and discussion sections accordingly, and the specific changes can be found in the revised manuscript.

Point 5:Discussion needs to be shortened by parts not related to the Results.

Response 5: We have accepted your comments and have reorganized the results and discussion sections of the paper, and the specific changes can be found in the reworked manuscript.

Point 6: The way the data was collected and compiled does not support the concluding postulate.

Response 6: The Hengduan Mountains, which have a complex topography, is located in a large corridor of the Central Asian migratory flyway,so the lakes along the Hengduan Mountains are important for waterbirds migration. The existing researches on lakes in Hengduan Mountain area are mostly concentrated in the central and southern part of the Mountains, which proves that many lakes are wintering grounds for migrating birds. With the increase of altitude in the north, the ecological functions of some lakes become migratory resting places. We wonder whether the ecological functions of lakes will change more with the further elevation. With this question, we conducted the seasonal birds' survey in Mangcuo Lake. We found that there were no wintering waterbirds in Mangcuo Lake, and relatively more waterbird species and numbers in the spring and autumn migratory seasons compared with some geese and ducks and grebes breeding here in summer. Therefore, we propose that Mangcuo Lake act as a breeding site for some waterbirds and furthermore a very important stopover for migrating birds in the alpine region of the Hengduan Mountains.

Minor points

Point 7: word inappropriate for scientific publishing

Original content:Resting places provide food and habitat for migratory birds and therefore play an important role in facilitating the birds' completing their migratory journey.

Response 7:Accepted, we modified this sentence into “The migration routes are usually composed of fixed wintering grounds, stopovers, and breeding grounds. The stopovers provide food and habitat for migratory birds and therefore play an important role in facilitating the birds' completing their migratory”.

Point 8: Summary data has no value. How many per year/season?

Original content:We conducted fieldwork for the seasonal birds' survey in Mangcuo Lake during 2019–2020. A total of 6,109 birds belonging to 20 species were recorded.

Response 8:Accepted, and we revised the sentenced as ”A total of 6,109 birds belonging to 20 species were recorded. In spring, 15 species of waterbirds with 4,210 individuals were observed; both summer and autumn had 9 species with 801 and 1,098 individuals respectively; no waterbirds were observed in winter”.

Point 9: a life strategy, not a behavioral one

Original content:Migration is an important behavioral strategy in birds, which adapt to their natural environment.

Response 9:Accepted and revised.

Point 10: has no connection to migrations

Original content:The migration routes are usually composed of fixed wintering grounds, 45

stopovers, and breeding grounds.

Response 10:Sorry we are not clear for this comment.

Point 11: what is the depth?

Original content:The water body of the lake spans 18 km2 in total, with a maxi-mum depth.

Response 11:Thanks the comment, we have added in the text that the maximum depth of Mangcuo Lake is 21 m.

Point 12: This should be described in detail: how many counts, in what seasons, how the dates varied from year to year, how many people conducted the censuses, and other details needed to compile the results

Original content: We carried out the zonal direct count method for the waterbird survey in all four seasons from October 2019 to July 2020.

Response 12:We have reorganized this section into: From November 2019 to July 2020, four field surveys of the waterbird community of Mangcuo Lake were conducted in all seasons using the zonal direct count method. Surveys were conducted in November 2019 (autumn), December 2019 (winter), April 2020 (spring), and July 2020 (summer), once in each season. The surveys were carried out in two teams of 2-3 persons each under clear weather conditions with little fog and no strong wind. Two sample areas were set up according to the topography, openness, and traffic conditions of the lake area. Sample area 1 was located in the transition area between swampy wetland and lake in the northeast of the lake area, and sample area 2 was located in a single lake habitat in the southeast. We used a Leica 82 mm telescope to observe the birds, positioned used GPS, and recorded data on waterbird species, numbers and habitat types. Smaller groups of birds were counted directly, and larger groups were estimated based on the density of birds within different visual field patches and the number of patches.

Point 13: How many, on what dates?

Original content: The field survey

Response 13:Thanks to your comments, This part of the reorganization is included in the Point 12 response.

Point 14:A complete lack of statistical methods necessary to compare variation in abundance and calculated indicators from year to year/season. It is necessary to use tests, such as Anova or Kruskal-Wallis

Response 14:We received your comments and analyzed the diversity of waterbirds in different habitats at the two observation sites. Shapiro test was used to test the normality of the data. If they conformed to normal distribution, one-way ANOVA was used for analysis; if they did not conform to normal distribution, Kruskal Wallis H was used for analysis, or the data were transformed into species presence/absence data, i.e., species detected at the sample sites were recorded as 1 and those not detected were recorded as 0, and binomial analysis was performed. These methods were used to test for seasonal and annual variability among waterbirds in different habitats. The data were analyzed using R 4.2.1 software with a significance criterion of P < 0. 05.

Point 15:The total sum of birds from several years has no value. It is impossible to make rational inferences on this basis. The Authors should rearrange the raw results from the counts in years/seasons, only then calculate the ratios, and compare the statistical significance of the results so compiled.

Response 15:Received your comments, we have reorganized this part of the content, Table 1 is not very meaningful to delete, 3.1 results part of the reorganization as The four-season survey recorded 6,109 individuals in total, belonging to 20 species in 6 orders and 8 families The Black-necked Grebe(I=63%), Great Crested Grebe (I=10%), Bar-headed Goose(I=11%) and Ruddy Shelduck(I=9%) were the dominant species in the waterbird community. Gadwall (Mareca strepera), Tufted Duck (Aythya fuligula), and Eurasian Coot (Fulica atra) are common species in the waterbird community of the region. The number of rare or occasional species is relatively high, with 13 species, accounting for 65% of the total number of waterbird species, mainly ducks, storks and herons. Among the 20 species of waterbirds recorded, 11 species were migrants (55%) and nine species were summer visitors (45%), no waterbird was recorded in winter, indicating that migrants are the majority of waterbirds in Mangcuo Lake .

Point 16: Table 1 —— Number(how many per year/season?)

Response 16:The raw data for each season, we collated to Appendix A.

Point 17: No raw data available 

Response 17:We add the raw data to Appendix A.

Point 18: The lowest indices of diversity and evenness were recorded in spring with 0.594 and 0.219 (Were the differences statistically significant?)

Response 18:We received your comment and apologize for not comparing significant differences for this part of the data

Point 19: Greater values of beta diversity show greater similarity, while smaller values indicate less similarity.(This sentence should be in Methods)

Response 19:Thank you for your comments, this sentence we have put in 2.2. Method.

Point 20: It is not clear.This section should be included in the discussion

Original content:3.3 Location Distance

Response 20:We have received your comments and after discussion, we think this section should be better placed in the results. In the discussion section, we have discussed the changes in the ecological functions of lakes for migrating waterbird at different elevations.

Point 21: Title too long, term travelers is not properly to scientific papers

Original content:4.1 The bird composition of Mangcuo Lake is dominated by traveling birds.

Response 21:Thanks to your comment, we have revised this title to read: The bird composition is dominated by migratory birds

Point 22: It is not clear, please explain 

Original content:Birds respond to seasonal fluctuations in their environment by migrating

Response 22:Thanks for your comments, we are not accurate representation of this part of the content, after reorganization, the deletion of it.

Point 23: title too long

Original content:4.2 Mangcuo Lake has the characteristics of a typical migratory stopover site.

Response 23:Thanks to your comment, we have revised this title to read: Mangcuo Lake is migratory stopover site 

Point 24:——Lack of connection with Results

Original content:A series of wetlands along the migratory route of migratory birds in the Hengduan Mountain system have been assigned different ecological functions. Among them, Jianhu, Qionghai, and Erhai, which are at lower elevations, mainly provide important wintering grounds for wintering waterbirds (especially geese and ducks) . Napa Sea, Lashihai, Lugu Lake, and Mangcuo Lake, which are located at higher altitudes, are both important migratory resting places for waterbird and important wetlands for waterbird wintering grounds (or breeding grounds) . As internationally important wetlands, both Napa Sea and Lashihai are shallow lakes with high similarity in species composition, providing overwintering sites and stopover sites for wading birds and shallow-water wading birds. Lugu Lake, an upland deep-water lake, is mostly utilized by deep-water species of wading birds for resting and overwintering.

Response 24:Having received your comments, we have reorganized this section as follows:Mangcuo Lake is far away from all these Hengduan mountain lake groups above, with a rather different eco-climate. Mangcuo Lake is located in the alpine zone, surrounded by mountains, and is the only alpine freshwater lake in the Hengduan Mountains above 4,000 m with an area of more than 10 km2 [19]. There are also differences in the ecological functions of migratory migratory birds between the Mangcuo Lake and the Hengduan Lake Group. Mangcuo Lake is a seasonal lake on the plateau. In winter, due to the cold climate, the lake surface freezes from December to March, and there are no overwintering waterfowl. But from April to November every year, there are suitable habitats for waterbirds such as meadows, shallow water marshes, and bright water areas. A small number of waterbirds breed here in summer, and most of them migrate through the area mainly in spring and autumn. Mangcuo Lake is a breeding site for some migrating waterbirds, and its main ecological function is to provide a stopover for migrating waterbirds from the southern wintering wetlands.

Point 25: Desrcibed dependence are not clear.

Original content:Due to the high altitude, harsh temperature, and absence of wintering waterbirds on the frozen lake surface in winter, the lake's primary serves as a breeding ground

Response 25:We have received your comments and have reorganized this section: In the migratory pathway of migratory birds in the Hengduan Mountain system, a series of wetlands are given different ecological functions, and with the elevation, the ecological functions of the lakes in the Hengduan Plateau supporting the life history of migratory waterbirds are gradually transformed about three levels, the first level of Qionghai, Chenghai, Erhai and Jianhu at an elevation of 1,500-2,200 m is the most important ecological function of providing important wintering grounds for wintering waterbirds (especially geese and ducks) [20-23]. The second layer of Lashihai, Lugu Lake and Napa Sea at 2,400-3,300 m above sea level are not only wintering wetlands for migratory birds but also important stopover sites [24-26]. The third layer, Mangcuo Lake, at an altitude of 4,000 m above sea level, provides a breeding ground for some migratory waterbirds in summer and a stopover for migratory waterbirds in spring and autumn.

Point 26: it schould be removed

Original content:4.3 The protection value of Mangcuo is underestimated

Response 26:Thank you for your comments, we have modified the text accordingly.

Point 27: What does in meant?

Original content:most dominant alpine lake

Response 27:Thank you for your comments, this sentence is not formulated accurately enough, rewrite it as Mangcuo Lake is the important alpine lake in the northwestern part of the Hengduan Mountains.

Point 28: The results presented in this paper do not justify such a postulate. Perhaps it is a matter of their poor elaboration and presentation

Original content:We advocate for the formation of a higher-quality nature reserve or national park in the region, to further balance the development and protection, scientific planning, and  coordinated development, so that to sustain the important ecological role of Mangcuo Lake role in supporting the migration of birds.

Response 28:Our survey of Mangcuo Lake revealed that it provides an important intermediate migration site for migrating waterbirds in the alpine region.we founded that the area was also active with Lammergeier(Gypaetus barbatus), Himalayan vultures(Gyps himalayensis), Tibetan antelope(Procapra picticaudata), Tibetan foxes(Vulpes ferrilata), wolves(Canis lupus) and other rare and endangered wildlife.Our findings, hopefully, will provide a reference direction for the future conservation and management of Mangcuo Lake.

Point 29: How many in each season, what was the variation between seasons?

Original content:Conclusions(1)The four-season survey recorded 6,109 individuals in total,belonging to 20 species in 6 orders and 8 families. In winter, no waterbird were recorded. The bird composition of Mangcuo Lake is dominated by traveling birds.The dominant species of the lake's waterbird community varied with the season.

Response 29:Thank you for your comments, we have modified the text accordingly. Revised to: (1)The four-season survey recorded 6,109 individuals in total, belonging to 20 species in 6 orders and 8 families. The waterbird community of Mangcuo Lake had large differences in different seasons, with migratory birds accounting for 87% of the total number of birds in spring and autumn and no waterbirds in winter. All the waterbirds staying in Mangcuo Lake were migratory birds, and geese, ducks, and grebes were the main migratory waterbirds in Mangcuo Lake.

Reviewer 2 Report

I have made corrections in track changes. Please see the attachment.

Author Response

Response to Reviewer 2 Comments

Point 1: mention some common and scientific names of some plant species

Original content: The vegetation around the Mangcuo Lake is dominated with alpine subarctic shrub meadow that provides food and habitats for birds and other highland wildlife.

Response 1: Your comments are very helpful. We add to the 2.1 study area: Mangcuo Lake is a typical highland wetland ecosystem, the vegetation is dominated with alpine subarctic shrub meadow, the meadow vegetation includes Kobresia pygmaea, Polygonum macrophyllum and Blysmus sinocompressus; the aquatic vegetation is Potamogeton pectinatus of the family Potamogetonaceae; the marsh vegetation is Triglochin maritimum and Hippuris vulgaris.

Point 2: mention some important names of cities, lakes, and coordinates on the margin so readers can know where this lake is located.

Original content: Figure 1. The location of Mangcuo Lake and the real view of the four seasons.

Response 2: Thanks your comments. We have marked important cities and rivers on the redrawn map so that readers can better understand the location of Mangcuo Lake.

Point 3: Give names of surrounding important sites. What are the two white maps in the lake? Are they terrestrial area? Do they have names?

Original content: Figure 2. Observation sites and zoning at Mangcuo Lake

Response 3: Thanks your comments. We have marked the cities around Mangcuo Lake in the redrawn map. The two white maps in the lake are the island in the heart of Mangcuo Lake, the area of land, which is not specifically named as far as we know.

Point 4: Note: Write scientific name only once and later use common names.

Response 4: Thanks your comments. This was a mistake in our writing and has been corrected accordingly in the paper.

Point 5: This table does not make any sense. It will be more useful to give names of birds in each family.

Original content: Table 1-Table 1. Classification and composition of waterbirds in Mangcuo Lake

Response 5:We have discussed and adopted your comments. The original Table 1 has been deleted and the bird information for each family has been added to Appendix A.

Point 6: Delete the yellow portion as it makes no sense.

Original content: Table 2. Seasonal dynamics of bird diversity in Mangcuo Lake

Response 6:We have used your comments to trim the annual data from Table 2.

Point 7: You have not mentioned which duck breeds in Mangcuo lake.

Original content: For birds migrating through the alpine region of the Hengduan Mountain chain, Mangcuo Lake is of enormous importance as a nesting place for some ducks and as a stopover for a large number of migratory passengers.

Response 7:Your comments are much appreciated and we have made the following changes in the corresponding section of the paper: Through the four-season survey of waterbirds in Mangcuo Lake, it was found that there were no wintering waterbirds in Mangcuo Lake, and there were relatively more waterbird species and numbers in the spring and autumn migratory seasons, and some geese and ducks and grebes breeding here in summer. The ecological functions of Mangcuo Lake and other lakes in the Hengduan Mountains are also different. For birds active in the alpine region of the Hengduan Mountains, Mangcuo Lake is both a breeding site for some waterbirds and a stopover for migrating birds(Table 4).

Point 8: Can you show these lakes in the map, in relation to Mangcuo Lake

Original content: Table 4. Relative position of Mangcuo Lake to the lake group

Response 8:After discussion, we strongly agree with your comments and have added a map of the relative location of Mangcuo Lake to the Hengduan Lake Group in the 2.1 study area section.

Point 9: How can you say this? There are wonderful studies on the Black-necked cranes in Tibet/China.

Original content: The documented migration route of the Black-necked Grebe is to migrate north from its southern overwintering grounds to breed in Inner Mongolia, northeast China, and northern Xinjiang in April each year, and to over- winter in the southeast and south China coasts and southwest China starting in late October[30]. There are no migratory and breeding studies of the Black-necked Grebe in highland regions.

Response 9:We received your comment and have made the following changes in the text:The documented migration route of the Black-necked Grebe is to migrate north from its southern overwintering grounds to breed in Inner Mongolia, northeast China, and northern Xinjiang in April each year, and to overwinter in the southeast and south China coasts and southwest China starting in late October[30]. In our study, 3660 Black-necked Grebes were recorded in Mangcuo Lake in April 2020, while 200 breeding individuals were subsequently observed in July. These results indicate that the population of Black-necked Grebes found here is moving north from the wintering wetland along the Hengduan Mountain in April, and after overstepping at Mangcuo Lake, the majority of individuals continue to go further north, while a small number of pairs breed in Mangcuo Lake.There are no relevant literature reports on the migration studies of black-necked Grebes in Tibet, China, and Mangcuo Lake is a good study area.

Finally, we would like to thank you for your very detailed work on the grammar and wording of the article to make it more precise, and we have revised this part of the article accordingly.

Reviewer 3 Report

The manuscript entitled “An important alpine breeding and stopover site along the cen

tral Asian flyway: evidence of bird surveys in Mangcuo Lake, Hengduan Mountains” investigated the waterbird community in Mangcuo Lake, Hengduan Mountains in the stopover site along Asian–Australasian flyway, which provide fundamental waterbird dataset in this region. Although this study only conducted four field surveys in four seasons (and no waterbird individual was found in winter), it could be the reference for other studies along the Flyway. And the manuscript contained several problems in writing, statistic analysis and logic, I provided some line comments.

Main concerns

(1) Data from four field surveys is not enough prove the importance of this region although some protected species at risk were observed during surveys. The site that was identify as important habitat/site should be based on multiple criterion. For example, abundance of some species in this site as a proportion of the flyway. You can refer to the definition of important or key habitat of waterbird to re-analysis the dataset (Xia Shaoxia et al, 2017, Biological Conservation).

(2) Because of only one field survey in each season, no statistic analysis was used in this study. You can compare waterbird diversity in two observation points and habitat types. And you should discuss the reasons why only four surveys in four seasons.

(3) It was valuable that you collected the relative position of Mangcuo Lake to the lake group. However, the readers prefer to see these lakes clearly on the map. I suggest author can mark the location of lakes on map through the flyway.

Minor points

Line 13, Please be careful in writting. Only four surveys can not be enough to define this site as important habitat. It should be changed as “Although the area of Mangcuo Lake is small, field survey showed some protected species at risk occurred in Mangcuo Lake……”.

Line 23-25, Please re-write this sentence.

Line 26-28, Please re-write this sentence. You should describe the result of your study clearly instead of the analysis based on the results.

Line 41, Please add “as” after the “considered”.

Line 49-50, As far as I know, many studies explore the effects of wetlands on migratory waterbird at stopover site. Please re-write this sentence.

Line 58-61, Please re-write this sentence. The purpose of your study is not to understand the movement of waterbird, but study the bird community at one site that was not identified as the stopover site along the flyway

Line 62, The title of “ Study Area and Methods” should belong to the second section.

Line 66, There should be spaces between units and figure.

Line 77-78, Please re-write this sentence. What work did you do? What method did you use? How many times?

Line 81, Please add space between units and figure.

Line 82, How many habitat types in this lake? And why did you not compare differences of bird community between habitat types?

Line 139, It should be changed as “no waterbird was recorded”.

Line 140-141, Please re-write this sentence. Because no statistic analysis was sued, it should not be described as “significantly higher”.

Line 173, Please mark the location of lake group on map through the flyway.

Line 285, Chinese characters should not appear.

Author Response

Point 1: Data from four field surveys is not enough prove the importance of this region although some protected species at risk were observed during surveys. The site that was identify as important habitat/site should be based on multiple criterion. For example, abundance of some species in this site as a proportion of the flyway. You can refer to the definition of important or key habitat of waterbird to re-analysis the dataset (Xia Shaoxia et al, 2017, Biological Conservation).

Response 1: Thanks for your Comment, we believe that Mangcuo Lake is important in three aspects. First, its unique position in the Hengduan Mountain lake group is about 200km away from the nearest lake, which makes it irreplaceable as a migration stop. The second is that it provides a choice of breeding habitat for many waterfowl and other animals such as , a useful buffer zone when bird migration is affected by other factors, such as extreme weather. Third, from the historical survey, Mangcuo Lake has rarely been paid attention to by scientists. Although a county-level protection zone of Mangkam County has been established here, it is difficult to support its unique ecological functions and protection needs. We hope that relevant departments can pay more attention to it.

We agree that “The site that was identify as important habitat/site should be based on multiple criterion”. But Mangcuo Lake is not suitable for Xia(2017)’s method, we have calculated the species recorded in the survey of Mangcuo Lake by referring to the global number of species in the IUCN Red List, but none of them reached 1% of the global number. For example, the total number of black-necked grebe is 3,860, accounting for about 0.1% of the global total, the total number of Great Crested Grebe is 638, accounting for about 0.07% of the global population, and the black-necked crane is 13, accounting for about 0.13% of the global population.The area of Mangcuo Lake is only 18 Km2, the altitude is over 4,000 m, so relatively few waterbirds were recorded, with 6109 birds of 20 species.

Point 2: Because of only one field survey in each season, no statistic analysis was used in this study. You can compare waterbird diversity in two observation points and habitat types. And you should discuss the reasons why only four surveys in four seasons.

Response 2: Thank you for your comments. We added a comparative analysis of the diversity of waterbirds in different habitat types at the two observation sites to our paper based on your suggestion, and found that there were significant differences in the distribution of birds in the two different habitats, with habitat complexity being an important influencing factor. Reasons for only four surveys in four seasons:1. Mangcuo Lake is located in a precarious and remote location, often landslides, mudslides and other disasters, the traffic is very inconvenient, which takes more than a week to travel to and from Lhasa to the lake(>1,200km). 2.It is very difficult for us to apply for special projects and funds to carry out research on Mangcuo Lake, Mangcuo Lake had no bird data available during the first comprehensive scientific expedition to the Qinghai-Tibet Plateau in the 1960s and 1980s, and no relevant studies have been conducted in the decades since then. 3.Mangcuo Lake is sparsely populated and the logistics are very rudimentary, and there is no field station nearby to support our continuous monitoring work.

Point 3: It was valuable that you collected the relative position of Mangcuo Lake to the lake group. However, the readers prefer to see these lakes clearly on the map. I suggest author can mark the location of lakes on map through the flyway.

Response 3: Thank you for your affirmation. We accepts your comments and add images of the relative positions of Mangcuo Lake and the Hengduan Lake Group in 3.4.

We believe that the lake group of Hengduan Mountain is hierarchical, and the relatively low altitude areas are all wintering areas for waterbirds. With the increase of altitude, some other lakes begin to function as a stop-over place, and Mangcao Lake is the southernmost breeding place and also a stopping place during migration.

Point 4: Line 13 (Although the area of Mangcuo Lake is small, it provides important habitat for some of its protected species at risk, and its conservation value is underestimated.), Please be careful in writting. Only four surveys can not be enough to define this site as important habitat. It should be changed as “Although the area of Mangcuo Lake is small, field survey showed some protected species at risk occurred in Mangcuo Lake……”.

Response 4: Thanks for your Comments. We also feel that this part is not written precisely enough, so we are rewriting the simple summary.

The original version of Simple Summary: In this study, four waterbird surveys were conducted at Mangcuo Lake in different seasons, and it was found that Mangcuo Lake not only provides an extremely important migratory resting place for waterbirds along the migration in the alpine region, but also is a breeding site for some birds. Although the area of Mangcuo Lake is small, it provides important habitat for some of its protected species at risk, and its conservation value is underestimated. We call for the establishment of a higher-ranking nature reserve or national park in the area, so that Mangcuo Lake can play a greater role in supporting the migration of birds.

Revised Simple Summary: The Hengduan Mountain area in southwest China is an important migratory channel for migratory birds in Central Asia, and the lakes along the way have different ecological functions for the life histories of waterbirds. In order to determine the ecological functions of high-altitude lakes, we selected Mangcuo Lake in Markam County to observe the waterbird community in four seasons and found that Mangcuo Lake provides breeding habitat for 9 species of birds and also supports greater numbers of birds migrating in the spring and autumn. In view of the role of Mangcuo Lake in the migration route, we propose to raise the protection level of Mangcuo Lake.

Point 5: Line 23-25(Seasonal changes in the waterbirds' species composition are varied among seasons, with significantly higher species numbers and population abundance in the migratory season than the breeding season.), Please re-write this sentence.

Response 5: Accepted. We rewrite this section: In spring, 15 species of waterbirds with 4,210 individuals were observed; both summer and autumn had 9 species with 801 and 1,098 individuals respectively; no waterbirds were observed in winter. The survey found that waterbirds in Mangcuo Lake had large differences in species and numbers in the transition zone between the marsh wetland and the lake and in the lake habitat, and that habitat complexity was an important factor affecting the number and distribution of birds(X2=14.405, P=0.000).

Point 6: Line 26-28(By comparing with other lakes in Hengduan Mountain, we found that Mangcuo Lake not only provides a very important stopover site for migratory waterbirds in the alpine area but also serves as a rare breeding ground for some birds.), Please re-write this sentence. You should describe the result of your study clearly instead of the analysis based on the results.

Response 6: Accepted. We rewrite this section:By comparing with other lakes in Hengduan Mountain, We found that the ecological functions of the plateau lakes in the Hengduan Mountains to support the life histories of migrating waterbirds are gradually transforming as the altitude rises, and can be divided into approximately three levels, with the first level of Qionghai, Chenghai, Erhai and Jianhu at altitudes of 1,500-2,200 m being the most important ecological function of providing wintering grounds for migrating birds. The second layer at an altitude of 2,400-3,300 m, Lashihai, Lugu Lake and Napahai are not only wintering wetlands for migratory birds but also important stopover sites. The third layer of Mangcuo Lake, above 4,000 m above sea level, provides a breeding ground for some migratory waterbirds in summer and a migratory resting place for migratory waterbirds in spring and autumn.

Point 7: Line 41(The species diversity and population abundance of birds have long been considered bio-indicators of environmental change in wetlands[7]), Please add “as” after the “considered”.

Response 7: Thank you for your comments. We have modified the corresponding section in the text.

Point 8: Line 49-50(However, there is a lack of knowledge on the effects of wetlands at stopover sites on migratory waterbirds.), As far as I know, many studies explore the effects of wetlands on migratory waterbird at stopover site. Please re-write this sentence.

Response 8: We received your comments, reviewed the literature and decided to delete this section

Point 9: Line 58-61(To better understand the migratory movements of birds, we explored Mangcuo Lake, the largest lake wetland in the northwestern part of the Hengduan Mountains, and surveyed the waterbird community and its seasonal dynamics.), Please re-write this sentence. The purpose of your study is not to understand the movement of waterbird, but study the bird community at one site that was not identified as the stopover site along the flyway.

Response 9: We received your comments, reviewed the literature and discussed them, and rewrote this section as follows:Our purpose was to investigate the function of Mangcuo Lake's role in the Central Asia-India migration corridor, and we therefore carried out a four-season bird survey study of the region.

Point 10: Line 62(1. Study Area and Methods), The title of “ Study Area and Methods” should belong to the second section.

Response 10: Thanks for your comments, it was a mistake in our writing and has been revised accordingly in the paper.

Point 11:Line 66( The water body of the lake spans 18km2 in total, with a maxi-mum depth.), There should be spaces between units and figure.

Response 11: Thanks for your comments, it was a mistake in our writing and has been revised accordingly in the paper.

Point 12: Line 77-78(We carried out the zonal direct count method for the waterbird survey in all four seasons from October 2019 to July 2020. ), Please re-write this sentence. What work did you do? What method did you use? How many times?

Response 12: We received your comments and have rewritten this section:From November 2019 to July 2020, four field surveys of the waterbird community of Mangcuo Lake were conducted in all seasons using the zonal direct count method. Surveys were conducted in November 2019 (autumn), December 2019 (winter), April 2020 (spring), and July 2020 (summer), once in each season. The surveys were carried out in two teams of 2-3 persons each under clear weather conditions with little fog and no strong wind. Two sample areas were set up according to the topography, openness, and traffic conditions of the lake area. Sample area 1 was located in the transition area between swampy wetland and lake in the northeast of the lake area, and sample area 2 was located in a single lake habitat in the southeast.

Point 13: Line 81(We used a Leica 82mm telescope to observe the birds, identify species, and count the number of each species as well as record the habitat types.), Please add space between units and figure.

Response 13: Thanks for your comments, it was a mistake in our writing and has been revised accordingly in the paper.

Point 14: Line 82(The field survey ), How many habitat types in this lake? And why did you not compare differences of bird community between habitat types?

Response 14: Thank you for your comments,We added a comparative analysis of the diversity of waterbirds in different habitat types at the two observation sites to our paper based on your suggestion, and found that there were significant differences in the distribution of birds in the two different habitats, with habitat complexity being an important influencing factor. 

Point 15: Line 139(so no waterbird were recorded.), It should be changed as “no waterbird was recorded”.

Response 15: Thank you very much for your suggestion, we have modified the paper accordingly.

Point 16: Line 140-141(In terms of seasonal dynamics, the species number and abundance was significantly higher in the spring and autumn than in summer season), Please re-write this sentence. Because no statistic analysis was sued, it should not be described as “significantly higher”.

Response 16: We received your comments, We have rewritten this section :Among the four seasons, the highest number of waterbird species and individuals were recorded in spring. The number of waterbird species ranged from high to low in spring, summer, autumn and winter, and the number of individuals ranged from high to low in spring, autumn, summer and winter.

Point 17: Line 173(Table 4. Relative position of Mangcuo Lake to the lake group ), Please mark the location of lake group on map through the flyway.

Response 17: We received your comments, We add images of the relative positions of Mangcuo Lake and the Hengduan Lake Group in 2. Study Area and Methods.

Point 18: Line 285(Appendix A: List of birds of Mangcuo Lake), Chinese characters should not appear.

Response 18: Accepted.We have revised the paper accordingly.

Reviewer 4 Report

The authors surveyed birds in Mangcuo Lake during 2019-2020 and evaluated the importance of the lake for bird accordingly. The paper is well-written and I have some minor comments on it.

1.       Please provide more detailed information on the background of Mangcuo Lake in the third paragraph of the Introduction. Does this area have a certain research basis? If there is a research basis, it is suggested to supplement the previous research results; if not, it is suggested to emphasize the originality of relevant work. 

2.       Some missing elevation information(Line 2, 98°47'10"~98°53'41"E, 29°27'2"~29°38'26"N, m.a.s.l) needs to be provided in 2.1 study area. 

3.       There is a slight disconnect between 3.1. Waterbird species composition and 3.2. Seasonal dynamics. Seasonal community composition is the result of more detailed investigations, but they are not presented, so it is suggested to add spring, summer and autumn community composition figures. 

4.  I suggest that the relative position of different lakes in Hengduan Mountain should be represented graphically in 3.3 Location Distance, and the shape outline of each lake should be drawn in the table.

Author Response

Response to Reviewer 4 Comments

Point 1: Please provide more detailed information on the background of Mangcuo Lake in the third paragraph of the Introduction. Does this area have a certain research basis? If there is a research basis, it is suggested to supplement the previous research results; if not, it is suggested to emphasize the originality of relevant work.

Response 1: Thank you for this very insightful comment. We've added background information about Mangcuo Lake in 1. Introduction: “The Mangcuo Lake, located in the northern part of the Hengduan Mountains at an altitude of over 4000 m, During the first comprehensive scientific expedition to the Tibetan Plateau in the 1960s and 1980s, the location of Mangcuo Lake was remote and precarious, the roads were not cleared, and no bird data were available for the area. In the decades since, there have been no reports on the status of bird resources in the area. Our purpose was to investigate the function of Mangcuo Lake's role in the Central Asia-India migration corridor, and we therefore carried out a four-season bird survey study of the region.”The changes have been highlighted and you can see them in the word document Revised version with traces of modification.

Point 2: Some missing elevation information(Line 2, 98°47'10"~98°53'41"E, 29°27'2"~29°38'26"N, m.a.s.l) needs to be provided in 2.1 study area.

Response 2: I'm sorry, this was a mistake in our writing, thanks for your comments. We made changes in 2.1. Study area: Mangcuo Lake is located in Markam County, Qamdo City, Tibet Autonomous Region, China, which is located in the southeast of Markam County and belongs to the Jinsha River Basin(98°47'10"~98°53'41"E, 29°27'2"~29°38'26"N, 4313 m.a.s.l). The changes have been highlighted and you can see them in the word document Revised version with traces of modification.

Point 3: There is a slight disconnect between 3.1. Waterbird species composition and 3.2. Seasonal dynamics. Seasonal community composition is the result of more detailed investigations, but they are not presented, so it is suggested to add spring, summer and autumn community composition figures.

Response 3: We appreciate your comments, and we have reorganized the results in sections 3.1 and 3.2, which can be found in the reworked manuscript. The changes have been highlighted and you can see them in the word document Revised version with traces of modification.

Point 4: I suggest that the relative position of different lakes in Hengduan Mountain should be represented graphically in 3.3 Location Distance, and the shape outline of each lake should be drawn in the table.

Response 4: We accept your Comment that the map can better illustrate the geographical relationship between Mangcuo Lake and the lake group, We have added maps of the relative locations of Mangcuo Lake and the Hengduan Lake Complex at 3.4. The shape outline of each lake has been drawn in the table.

Round 2

Reviewer 1 Report

The paper has been revised, but there is still a section of discussion in the results chapter. Introduction and discussion need to be significantly shortened, conclusions are based in part on literature and not on the authors' research, language needs improvement. 

Author Response

Response to Reviewer 1 Comments

The paper has been revised, but there is still a section of discussion in the results chapter. Introduction and discussion need to be significantly shortened, conclusions are based in part on literature and not on the authors' research, language needs improvement. 

Response Overall:We thank the reviewers for pointing out the problems and for the meticulous revisions, which benefited the author team all. We have integrated the two parts by placing the original result 3.4 Transverse Mountain Lake Group in Discussion 4.2, and shortened introduction and discussion appropriately, we also improve the language. We still have a little different ideas about whether there is literature data in the conclusion. Considering that the study of this paper alone cannot support the results of changes in the ecological functions of lakes along the Hengduan Mountains, we tend to describe the existing results properly and then draw the conclusion of the ecological functions of Mangcuo Lake.

Point 1:In general, the abstract is too long and lacks substance

Response 1:Accepted. In the results section of the abstract, we modified it to read: We recorded a total of 6,109 birds of 20 species of waterbird, including 20 species of migratory waterbird, accounting for 100% of all bird species. The diversity richness of waterbirds in Mangcuo Lake is shown as spring > autumn ≥ summer > winter, with no waterbirds in winter. The Black-necked Grebe (Podiceps nigricollis), Great Crested Grebe (Podiceps cristatus), Bar-headed Goose (Anser indicus) and Ruddy Shelduck (Tadorna ferruginea) were the dominant species in the waterbird community. The highest number of waterbird species and total individuals were found in the transition zone of the marsh wetlands and lakes, and the number of waterbird species differed significantly among habitats (X2 = 14.405, P = 0.000), with habitat complexity being an important factor influencing waterfowl abundance and distribution.

Point 2: The existing researches on lakes in Hengduan Mountain area are mostly concentrated in the central and southern part of the Mountains, which proves that many lakes are wintering grounds for migrating birds. With the increase of altitude in the north, the ecological functions of some lakes become migratory resting places

Response 2:Accepted in part. We propose to retain ”The existing researches on lakes in Hengduan Mountain area are mostly concentrated in the central and southern part of the Mountains, which proves that many lakes are wintering grounds for migrating birds”, We would like to emphasize that, in the context that the lakes in the central and southern Hengduan Mountains have been confirmed to be mostly migratory wintering grounds, how will the ecological function of Mangcuo Lake change with the increase of altitude.

Point 3:Wetlands are one of the most productive ecosystems on earth, providing habitat for many kinds of waterbirds[1]. It has been proven that wetland environment change can directly and indirectly affect waterbird communities. Habitat suitability serving as the primary determinant of waterbird population dispersal and distribution, as well as the population abundance of waterbird[2-4].Waterbirds are sensitive to changes in wetland environments because they depend on a number of wetlands along the migration route[5-6].

Response 3:We agree with the reviewer that this part is somewhat fragmented, and we also believe that it is necessary to properly explain the relationship between wetlands, habitats and migratory waterbirds. so we prefer this section to be modified to read: Wetlands are among the most productive ecosystems on Earth, and waterbirds are sensitive to changes in wetland environments because they depend on a number of wetlands along their migration routes [1-3], environmental changes can affect waterbird populations directly or indirectly, and wetland suitability is a major determinant of waterbird population dispersal and distribution [4-6].

Point 4:Bird survey

Original content: Survey method

Response 4:Accepted.

Point 5:It is no results, it is discussion

Original content: 3.4.Lake Group in the Hengduan Mountains

Response 5:Accepted. We put this section in the discussion in 4.2.

Point 6:Bird species composition

Original content:The bird composition is dominated by migratory birds

Response 6:Accepted.

Point 7: All the waterbirds staying in Mangcuo Lake were migratory birds, and geese, ducks, and grebes were the main migratory waterbirds in Mangcuo Lake.

Response 7:Accepted.

Point 8: and Mangcuo Lake has great potential to discover more migratory occasional species through long-term observation

Response 8:This part is to provide some basis for the subsequent protection of Mangcuo Lake.

Point 9: These results indicate that the population of Black-necked Grebes found here is moving north from the wintering wetland along the Hengduan Mountain in April, and after overstepping at Mangcuo Lake, the majority of individuals continue to go further north, while a small number of pairs breed in Mangcuo Lake.

Response 9:In this section, we would like to set the stage for a subsequent study on the migration of black-necked Grebes based on satellite tracking. We changed it to read: According to the results of the study, it can be assumed that the black-necked Grebes population at Mangcuo Lake is likely to migrate northward from its wintering wetlands in southwestern China in April, and after a stopover at Mangcuo Lake, the majority of individuals continue to go further north, while a small number of pairs breed in Mangcuo Lake.

Point 10: Mangcuo Lake is located in the alpine zone, surrounded by mountains, and is the only alpine freshwater lake in the Hengduan Mountains above 4,000 m with an area of more than 10 km2 [19]. There are also differences in the ecological functions of migratory migratory birds between the Mangcuo Lake and the Hengduan Lake Group. Mangcuo Lake is a seasonal lake on the plateau. In winter, due to the cold climate, the lake surface freezes from December to March, and there are no overwintering water 

fowl. But from April to November every year, there are suitable habitats for waterfowl such as meadows, shallow water marshes, and bright water areas.

Response 10:We have integrated the two parts by placing the original result 3.4 Transverse Mountain Lake Group in Discussion 4.2:Southwest China's wetlands, which are located in the Central Asian migration zone of the global migratory corridor and are key wintering grounds and stopovers for many wetland waterbirds, play an important role in the global migration of migratory birds[16]. There are numerous lakes in the Hengduan Mountains(Figure 3), such as Erhai, Chenghai, Lugu Lake, Qionghai, Napahai, Lashihai, and Jianhu, constituting recognized waterbird distribution habitats[19]. There are some differences between the ecoclimate of Mangcuo Lake and the Hengduan Lake Group. Mangcuo Lake is located in the alpine zone, surrounded by mountains on all sides, and is the only alpine freshwater lake with an area of more than 10 km2 above 4,000 m above sea level in the Hengduan Mountains[19], and it is far away from the lake group, and the distance from the nearest Napa Sea is more than 200 km in a straight line. Mangcuo Lake is a highland seasonal lake with a cold winter climate, with temperatures as low as -10°from December to March, when the lake surface freezes. The following April to November, the temperature warms up and the lake has habitat types such as meadow, shallow water marsh and open water surface. The existing researches on lakes in Hengduan Mountain area are mostly concentrated in the central and southern part of the Mountains, which proves that many lakes are wintering grounds for migrating birds. With the increase of altitude in the north, the ecological functions of some lakes become migratory resting places. The ecological functions of Mangcuo Lake and other lakes in the Hengduan Mountains to support migratory birds are also different through the four-season survey of waterbirds in Mangcao Lake(Table 4). There are no wintering waterbirds in Mangcuo Lake, but there are relatively more species and numbers of migratory waterbirds in spring and autumn, and some geese and grebes breed here in summer.

Point 11: In the migratory pathway of migratory birds in the Hengduan Mountain system, a series of wetlands are given different ecological functions, and with the elevation, the ecological functions of the lakes in the Hengduan Plateau supporting the life history of migratory waterbirds are gradually transformed about three levels, the first level of Qionghai, Chenghai, Erhai and Jianhu at an elevation of 1,500-2,200 m is the most important ecological function of providing important wintering grounds for wintering waterbirds  (especially geese and ducks) [20-23]. The second layer of Lashihai, Lugu Lake and Napa Sea at 2,400-3,300 m above sea level are not only wintering wetlands for migratory birds  but also important stopover sites [24-26]. The third layer,

Response 11:We received your comments, and after reviewing the literature and discussing it, this section is an important concluding part of the paper, and for now we would still like to keep this section.

Point 12: This title does not make sense 

Original content:4.3. The protection value of Mangcuo schould be removed

Response 12:The title of 3.4 should read: The protection value of Mangcuo should be improved. We have written this paper on two topics. On the one hand, we wonder whether the ecological functions of lakes will change more with the further elevation. On the other hand, based on the findings, we found that the conservation importance of this region is higher than the current conservation management approach, and we hope that this study will provide a feasible reference for biodiversity conservation in the alpine region of Markam.

Point 13: Although Mangcuo Lake is the important alpine lake in the northwestern part of the Hengduan Mountains[19], its significant value of migratory resting place for birds is underestimated. 

Response 13:We have rewritten this sentence as Mangcuo Lake is an important alpine lake in the northwest section of the Hengduan Mountains[19], but its conservation value in the alpine region has been underestimated.

Point 14: For example, in 2020, we recorded a Black Stork, the second record in the whole of Tibet[37].

Response 14:Thank you very much for your comment, because based on the fact that this study only confirmed the clear distribution of black storks in Tibet Autonomous Region(TAR) of China, we are inclined to rewrite the sentence as: The black stork recorded in Mangcao Lake in this study is one of the important arguments to confirm the distribution of black stork in TAR, as there is no documented information on the distribution of black stork in the TAR before.

Point 15: This information does not apply to the results  

Original content:In addition to the waterbird survey, we founded that the area was also active

with Lammergeier(Gypaetus barbatus), Himalayan vultures(Gyps himalayensis), Tibetan ante-

lope(Procapra picticaudata), Tibetan foxes(Vulpes ferrilata), wolves(Canis lupus) and other rare

and endangered wildlife.

Response 15:We found these protected beasts in our work conducting bird surveys, but they were not part of our findings. We took this part into consideration when discussing the conservation value of MangcuoLake and hoped that the conservation of this area would receive more attention.

Point 16: The authors studied birds only on Lake Mangcuo, so they have no basis for differentiating the levels of other lakes

Original content:We found that the ecological functions of the plateau lakes in the Hengduan 371

Mountains to support the life histories of migrating waterbirds are gradually transforming as the altitude rises, and can be divided into approximately three levels, with the first level of Qionghai, Chenghai, Erhai and Jianhu at altitudes of 1,500-2,200 m being the most important ecological function of providing wintering grounds for migrating birds. The second layer at an altitude of 2,400-3,300 m, Lashihai, Lugu Lake and Napahai are not only wintering wetlands for migratory birds but also important stopover sites.

Response 16:Thanks to your suggestion, because there has been some work around the lakes in the central and southern part of Hengduan Mountain(We have reviewed the literature related to the birds of the Hengduan Lake Complex and upload the relevant pdf documents in the reference section), and the ecological functions of those lakes have been divided into two categories: 1 Complete overwintering ground, 2 Overwintering ground and also a migratory stopover. Considering that the study of this paper alone cannot support the results of changes in the ecological functions of lakes along the Hengduan Mountains, we tend to describe the existing results properly and then draw the conclusion of the ecological functions of Mangcuo Lake.

Lake

Dominant species

References

Erhai

Common Moorhen,Eurasian Coot, Little Grebe, Great Crested Grebe, Eurasian Teal, Gadwall, Ferruginous Duck

Chenghai

Eurasian Coot, Black-headed Gull

Qionghai

Eurasian Coot, Black-headed Gull, Little Grebe, Common Moorhen, Great Cormorant, Little Egret, Gadwall, Eurasian Teal

Jianhu

Eurasian Teal, Eurasian Coot, Grey-headed Swamphen, Ruddy Shelduck, Gadwall, Tufted Duck, Common Pochard, Greylag Goose, Little Grebe, Great Crested Grebe, Common Moorhen, Northern Lapwing

Lashihai

Bar-headed Goose, Ruddy Shelduck, Eurasian Teal,Mallard, Red-crested Pochard, Common Pochard, Eurasian Coot

Lugu Lake

Eurasian Coot, Red-crested Pochard, Ruddy Shelduck, Greylag Goose, Common Pochard, Black-headed Gull

Napa Sea

Little Grebe, Bar-headed Goose, Ruddy Shelduck, Eastern Spotbilled Duck, Mallard, Black-necked Crane, Northern Lapwing

Point 17: surprising conclusion

Original content:The protection value of Mangcuo schould be removed

Response 17:The title of 3.4 should read: The protection value of Mangcuo should be improved. China's protected areas are divided into national, provincial, municipal and county levels, and Mangtso Lake Nature Reserve is currently a county-level reserve. According to the results of our survey, the conservation value of Mangcuo Lake is underestimated, and we hope that the conservation of this area will receive more attention and focus. Therefore, we suggest that Mangcuo Lake Reserve be upgraded to a provincial reserve to provide a direction for the subsequent protection and management of Mangcuo Lake.

Reviewer 3 Report

The manuscript entitled “Mangcuo Lake in Hengduan Mountains: An important alpine breeding and stopover site along Central Asian flyway, No. 2192923 has described the importance of Mangcuo Lake for the waterbird conservation along the Central Asian Flyway. The manuscript has been revised in Abstract, Introduction, method and discussion sections. The latest version is clear in writting, data analysis and logic, and I think this manuscript could be accept .

Author Response

Point 1 The manuscript entitled “Mangcuo Lake in Hengduan Mountains: An important alpine breeding and stopover site along Central Asian flyway, No. 2192923” has described the importance of Mangcuo Lake for the waterbird conservation along the Central Asian Flyway. The manuscript has been revised in Abstract, Introduction, method and discussion sections. The latest version is clear in writting, data analysis and logic, and I think this manuscript could be accept .

Response 1: Thanks for the reviewer's affirmation, we have refined some of the language, and also revised part of the  introduction and discussion.